# Short communication: On the potential use of materials with heterogeneously distributed parent and daughter isotopes as primary standards for non-U-Pb geochronological applications of laser ablation inductively coupled plasma mass spectrometry (LA-ICP-MS)

Daniil V. Popov[1,2]

[1] School of Environment, Earth and Ecosystem Sciences, The Open University, Walton Hall, Milton Keynes MK7 6AA, UK
[2] Scottish Universities Environmental Research Centre, Rankine Avenue, Scottish Enterprise Technology Park, East Kilbride G75 0QF, UK

*Correspondence to*: Daniil V. Popov (d.vs.popov@gmail.com)

**Abstract.** Many new geochronological applications of laser ablation inductively coupled plasma mass spectrometry (LA-ICP-MS) have been proposed in recent years. One of the problems associated with this rapid growth is the lack of chemically and isotopically homogeneous matrix-matched primary standards to control elemental fractionation during LA-ICP-MS analysis. In U-Pb geochronological applications of LA-ICP-MS this problem is often addressed by utilising matrix-matched primary

standards with variable chemical and isotopic compositions. Here I derive a set of equations to adopt this approach for non-U-Pb geochronological applications of LA-ICP-MS.

## 1 Introduction

The use of laser ablation inductively coupled plasma mass spectrometry (LA-ICP-MS) for in situ geochronology is growing rapidly, and recent years have seen this technique being applied to many new minerals and isotope systems. Examples include

in situ U-Pb dating of apatite (Chew et al., 2014, 2011), carbonates (Roberts et al., 2017; Li et al., 2014; Guillong et al., 2020) and epidote (Peverelli et al., 2021), Rb-Sr dating of micas (Hogmalm et al., 2017; Zack and Hogmalm, 2016), alkali feldspar (Bevan et al., 2021) and celadonite (Laureijs et al., 2021), Lu-Hf dating of garnet, apatite and xenotime (Simpson et al., 2021), and a new approach to Re-Os dating of molybdenite (Hogmalm et al., 2019). One important challenge associated with this rapid growth is the development of matrix-matched primary standards to correct for elemental fractionation during LA-ICP-

MS analysis. Ideally, primary standards should be chemically and isotopically homogeneous and isostructural to the analysed minerals. However, finding or synthesising such materials is not trivial. Therefore, recent studies relied on some alternative solutions, including the use (i) matrix-matched standards with variable chemical and isotopic composition (e.g. Chew et al., 2014) and (ii) nanoparticulate pressed powder tablets as substitutes for chemically and isotopically homogeneous matrix-matched standards (e.g. Hogmalm et al., 2017).

Matrix-matched primary standards with variable contents of parent and daughter isotopes are often used in U-Pb geochronological applications of LA-ICP-MS. Chew et al. (2014) proposed several approaches for dating common Pb-bearing phases, in which primary standards with variable contents of common Pb are used to characterise U-Pb fractionation. In all of these, individual primary standard analyses are corrected for common Pb before factors to correct for U-Pb fractionation are calculated from them, such that the latter step only relies on comparing the observed and expected $^{238}U/^{206}Pb_{radiogenic}$ ratios (as

opposed to using the $^{238}U/^{206}Pb_{total}$ ratios). The approaches differ in how the common Pb correction is introduced. This can be achieved by using $^{204}Pb$ or, assuming that no $^{232}Th$ is present, $^{208}Pb$ to estimate the amounts of common $^{206}Pb$ and calculate the $^{238}U/^{206}Pb_{radiogenic}$ ratios. Alternatively, straight tie-lines can be projected from an a priori estimate for the initial $^{207}Pb/^{206}Pb$ ratio through the acquired U-Pb data to the concordia in the Tera-Wasserburg space to calculate the $^{238}U/^{206}Pb_{radiogenic}$ ratios (Fig. 1a). A similar approach was adopted by Li et al. (2014), Roberts et al. (2017) and Guillong et al. (2020), who first used

chemically and isotopically homogeneous non-matrix-matched primary standards to correct for any drift in U-Pb fractionation during analytical sessions, and then used matrix-matched primary standards with variable contents of common Pb to calculate correction factors for matrix-dependent U-Pb fractionation. The latter was done by plotting multiple primary standard analyses in the Tera-Wasserburg diagram to fit a straight line through them and compare its observed and expected intercepts with the concordia.

Apparently, there are only two non-U-Pb geochronological applications of LA-ICP-MS where materials with variable contents of parent and daughter isotopes were essentially used as a primary standard. The first is the pioneering work on in situ Rb-Sr dating by Zack and Hogmalm (2016). These authors calculated what factor is needed to correct for Rb-Sr fractionation in one biotite sample with known age to obtain an isochron of that age and then applied it to other samples measured on the same day. Another is the work of Bevan et al. (2021), who performed Rb-Sr analysis of two alkali feldspar samples alongside and

then corrected the data for one of them using the Rb-Sr fractionation factors that were deduced by comparing the observed and expected isochron slopes for the other. Most of recent studies relied on using nanoparticulate pressed powder tablets as substitutes for chemically and isotopically homogeneous matrix-matched primary standards (Hogmalm et al., 2017, 2019; Olierook et al., 2020; Li et al., 2020; Tillberg et al., 2021). However, the ablation properties of nanoparticulate pressed powder tablets are different from those of single crystals, and while they perform better as primary standards compared to glasses, Rb-

Sr dates obtained by using them can be offset from the expected values by several % (mostly up to 4 %, occasionally up to 7 % in tests of Redaa et al., 2021). Therefore, the quest for matrix-matched standards remains open. With this communication I aim to highlight that the idea of using materials with heterogeneously distributed parent and daughter isotopes as primary standards may have been abandoned prematurely and provide a set of equations for doing so.

## 2 Proposed Solution

Presumably, one of the reasons why the idea of using primary standards with variable contents of parent and daughter isotopes was abandoned in non-U-Pb geochronological applications of LA-ICP-MS is the absence of a clear approach to calculate

factors for elemental fractionation correction and their uncertainties. Clearly, these factors can be estimated by adopting one of the approaches used for U-Pb dating. For example, they can be estimated from individual primary standard analyses by using parent to daughter isotope ratios that are corrected for the presence of the non-radiogenic component of the daughter isotope using a common isotope, which is analogous to the aforementioned use of the [204]Pb- and [208]Pb-based correction methods. Alternatively, they can be estimated by plotting two-point normal or inverse isochrons based on individual primary standard analyses and finding by what coefficients the measured elemental ratios need to be multiplied to bring the slopes of the apparent normal or inverse isochrons to the true values (Fig. 1b, c). Finally, these factors can be estimated by calculating and comparing the observed and expected intercepts with the horizontal axis in the inverse isochron diagram, which is analogous to the aforementioned use of the Tera-Wasserburg diagram (Fig. 1c). But how to calculate elemental fractionation correction factors in an efficient way? And how to estimate their uncertainties and propagate these to the date uncertainties? Below I derive equations that can be used to do so. I rely on the uncertainty propagation law (JCGM, 2008). However, as Pieter Vermeesch's review outlines, other approaches exist, such as the method of maximum likelihood.

**2.1 Normal Isochron Space**

In the normal isochron space, the true composition of a primary standard with heterogeneously distributed parent and daughter isotopes is given by Eq. (1):

$$Y = Y_0 + X(e^{\lambda t} - 1) \, , \tag{1}$$

where $Y$ is the daughter to common isotope ratio (e.g., $^{87}Sr/^{86}Sr$), $Y_0$ is the initial daughter to common isotope ratio (e.g., $^{87}Sr/^{86}Sr_0$), $X$ is the parent to common isotope ratio (e.g., $^{87}Rb/^{86}Sr$), $\lambda$ is the decay constant and $t$ is the age of the primary standard. The analysis of this primary standard by LA-ICP-MS yields some proxies for the true $Y$ and $X$ values, which are the measured $y$ and $x$ values, respectively. It is generally presumed that any difference between $Y$ and $y$ is a result of mass dependent fraction that can be corrected for independently of analysing the primary standard in question (e.g. Chew et al., 2011; Li et al., 2014; Hogmalm et al., 2017; Redaa et al., 2021) or by acquiring data for pairs of non-radiogenic isotopes while analysing this primary standard (Bevan et al., 2021). Thus, $Y$ can be assumed to be equal to the product of $y$ and the obtained mass fractionation correction factor $l$. Any difference in $X$ and $x$ can be described in terms of elemental fractionation, which is heavily dependent on the matrix properties and ablation conditions, and which is being characterised by analysing the primary standard in question (e.g. Chew et al., 2011; Li et al., 2014; Hogmalm et al., 2017; Redaa et al., 2021). Thus, it can be assumed that $X$ is equal to the product of $x$ and the yet unknown factor $k$ needed to correct for elemental fractionation. With these assumptions Eq. (1) can be modified to make Eq. (2):

$$ly = Y_0 + kx(e^{\lambda t} - 1) \, , \tag{2}$$

from which it is possible to obtain an expression for $k$ given by Eq. (3):

$$k = \frac{ly - Y_0}{x(e^{\lambda t} - 1)} \, . \tag{3}$$

The uncertainty of thereby calculated $k$ can be estimated using Eq. (4):

$$\sigma_k^2 = \sigma_x^2 \left(\frac{-k}{x}\right)^2 + \sigma_y^2 \left(\frac{lk}{yl - Y_0}\right)^2 + 2\sigma_{x,y} \left(\frac{-lk^2}{x(yl - Y_0)}\right) + \sigma_t^2 \left(\frac{-\lambda k}{1 - e^{-\lambda t}}\right)^2 + \sigma_\lambda^2 \left(\frac{-tk}{1 - e^{-\lambda t}}\right)^2 +$$

$$+ 2\sigma_{\lambda,t} \left(\frac{t\lambda k^2}{(1 - e^{-\lambda t})^2}\right) + \sigma_l^2 \left(\frac{yk}{yl - Y_0}\right)^2 + \sigma_{Y_0}^2 \left(\frac{-k}{yl - Y_0}\right)^2 + 2\sigma_{Y_0,t} \left(\frac{\lambda x k^3 e^{\lambda t}}{(yl - Y_0)^2}\right), \tag{4}$$

where only the first three terms should be used to calculate the internal uncertainty ($\sigma_{k\,int}$), and the entire equation should be used to calculate the external uncertainty ($\sigma_{k\,ext}$). In principle, $\sigma_{Y_0,t}$ and $\sigma_{\lambda,t}$ should be different from zero if $t$ was determined using the same $Y_0$ and $\lambda$ as in the equations above (i.e., the age of the primary standard is not determined using some other method). Provided that this is the case, $\sigma_{Y_0,t}$ and $\sigma_{\lambda,t}$ can be estimated using Eq. (5-6):

$$\sigma_{\lambda,t} = -\frac{t\sigma_\lambda^2}{\lambda} \, , \tag{5}$$

$$\sigma_{Y_0,t} = -\frac{\sigma_{Y_0}^2}{\lambda X^* e^{\lambda t}} = \frac{\sigma_{Y_0}^2 \eta^*}{\lambda e^{\lambda t}} \, , \tag{6}$$

where $X^*$ is the parent to common isotope ratio used to determine $t$ if it was determined from a single analysis, while $\eta^*$ is the partial derivative of the isochron slope with respect to $Y_0$ if $t$ was determined by fitting an isochron. I show in the Appendix how to calculate $\eta^*$ if the isochron was fitted by the method of York et al. (2004). Note that for well-characterised primary standards $\sigma_{Y_0,t}$ will most likely be negligibly small. It is also likely that the contribution from the uncertainties in $l$ and $Y_0$ to the uncertainty in $k$ will be negligibly small.

Repeated primary standard analyses will yield $k_1$ to $k_N$, for which the weighted mean value $k_{wm}$ can be obtained via Eq. (7):

$$k_{wm} = \frac{\sum_1^N k_i w_i}{\sum_1^N w_i} \, , \tag{7}$$

where $w_i = \sigma_{k_i int}^{-2}$.

The uncertainty of $k_{wm}$ is given by Eq. (8):

$$\sigma_{k_{wm}}^2 = \frac{1}{\sum_1^N w_i} + \sigma_\lambda^2 \left(\frac{tk_{wm}}{e^{-\lambda t} - 1}\right)^2 + \sigma_t^2 \left(\frac{\lambda k_{wm}}{e^{-\lambda t} - 1}\right)^2 + 2\sigma_{\lambda,t} \left(\frac{t\lambda k_{wm}^2}{(e^{-\lambda t} - 1)^2}\right) + \sigma_l^2 \left(\frac{k_{wm}}{l} + \frac{Y_0}{l(e^{\lambda t} - 1)\sum_1^N w_i} \sum_1^N \frac{w_i}{x_i}\right)^2 +$$

$$+ \sigma_{Y_0}^2 \left(\frac{-1}{(e^{\lambda t} - 1)\sum_1^N w_i} \sum_1^N \frac{w_i}{x_i}\right)^2 + 2\sigma_{Y_0,t} \left(\frac{\lambda k_{wm}}{(e^{\lambda t} + e^{-\lambda t} - 2)\sum_1^N w_i} \sum_1^N \frac{w_i}{x_i}\right), \tag{8}$$

where the first term gives the internal uncertainty ($\sigma_{k_{wm} int}$), while the entire equation gives the external uncertainty ($\sigma_{k_{wm} ext}$).

Following the same assumptions and notation as above, an analysis of an unknown yields $y_u$ that should be corrected for mass dependent fractionation to estimate the true value $Y_u$ and $x_u$ that should be corrected for elemental fractionation to estimate the true value $X_u$. The estimated true values and their uncertainties can be calculated using Eq. (9-12):

$$X_u = k_{wm} x_u \,, \tag{9}$$

$$Y_u = l y_u \,, \tag{10}$$

$$\sigma_{X_u}^2 = \sigma_{x_u}^2 k_{wm}^2 + \sigma_{k_{wm}}^2 x_u^2 \,, \tag{11}$$

$$\sigma_{Y_u}^2 = \sigma_{y_u}^2 l^2 + \sigma_l^2 y_u^2 \,. \tag{12}$$

The first terms in the latter two equations provide the internal uncertainties ($\sigma_{X_u int}$ and $\sigma_{Y_u int}$), while the entire equations provide the external uncertainties ($\sigma_{X_u ext}$ and $\sigma_{Y_u ext}$).

The covariance between $Y_u$ and $X_u$ is given by Eq. (13):

$$\sigma_{X_u,Y_u} = \sigma_{x_u,y_u} l k_{wm} + \sigma_l^2 y_u x_u \left( \frac{k_{wm}}{l} + \frac{Y_0}{l(e^{\lambda t}-1)\sum_1^N w_i} \sum_1^N \frac{w_i}{x_i} \right) \,, \tag{13}$$

where only the first term should be used to calculate the covariance related to the internal uncertainties ($\sigma_{X_u,Y_u int}$, such that $\rho_{X_u,Y_u int} = \sigma_{X_u,Y_u int} \sigma_{X_u int}^{-1} \sigma_{Y_u int}^{-1} = \rho_{x_u,y_u}$), while the entire equation should be used to calculate the covariance related to the external uncertainties ($\sigma_{X_u,Y_u ext}$, such that $\rho_{X_u,Y_u ext} = \sigma_{X_u,Y_u ext} \sigma_{X_u ext}^{-1} \sigma_{Y_u ext}^{-1}$). Note that all of the variables in the expression in brackets are related to the primary standard.

Eq. (14) can be used to calculate the date of the unknown $T_{spot}$ from $X_u$ and $Y_u$ obtained during one measurement and the independently estimated initial isotopic composition $Y_{0u}$:

$$T_{spot} = \frac{\ln\left(\frac{Y_u-Y_{0u}}{X_u}+1\right)}{\lambda} \,. \tag{14}$$

The uncertainty $T_{spot}$ is given by Eq. (15):

$$\sigma_{T_{spot}}^2 = \sigma_{X_u}^2 \left( \frac{Y_{0u}-Y_u}{X_u \lambda (Y_u-Y_{0u}+X_u)} \right)^2 + \sigma_{Y_u}^2 \left( \frac{1}{\lambda(Y_u-Y_{0u}+X_u)} \right)^2 + 2\sigma_{X_u,Y_u} \left( \frac{Y_{0u}-Y_u}{X_u \lambda^2 (Y_u-Y_{0u}+X_u)^2} \right) +$$
$$+ \sigma_{Y_{0u}}^2 \left( \frac{-1}{\lambda(Y_u-Y_{0u}+X_u)} \right)^2 + \sigma_\lambda^2 \left( \frac{-T_{spot}}{\lambda} \right)^2 \,, \tag{15}$$

where using the first three terms with $\sigma_{X_u int}$, $\sigma_{Y_u int}$ and $\sigma_{X_u,Y_u int}$ provides the internal uncertainty ($\sigma_{T_{spot}int}$), while using the entire equation with $\sigma_{X_u ext}$, $\sigma_{Y_u ext}$ and $\sigma_{X_u,Y_u ext}$ provides the external uncertainty ($\sigma_{T_{spot}ext}$). Note that $\sigma_{X_u,\lambda}$ is zero, and thus the associated covariance term is absent in this and following equations.

Multiple measurements of the same unknown within the same batch of analyses will give sets of $X_u$, $Y_u$, $\sigma_{X_u int}$, $\sigma_{Y_u int}$ and $\rho_{X_u, Y_u int}$, which can be used to fit a single isochron by the method of York et al. (2004), whether pinned or unpinned to $Y_{0u}$.

140 This procedure will yield the slope of the isochron $b$ and its internal uncertainty $\sigma_{b\,int}$. The external uncertainty of $b$ ($\sigma_{b\,ext}$) is given by Eq. (16), where $\sigma_{l,k_{wm}}$ is already taken into account, and all of the variables in the rightmost pair of brackets are related to the primary standard:

$$\sigma^2_{b\,ext} = \sigma^2_{b\,int} + \sigma^2_{k_{wm}} \left(\frac{-b}{k_{wm}}\right)^2 + \sigma^2_l \left(\frac{b}{l}\right)^2 \left(-1 - \frac{2Y_0}{k_{wm}(e^{\lambda t}-1)\sum_1^N w_i} \sum_1^N \frac{w_i}{x_i}\right). \tag{16}$$

Measurements of the same unknown that were obtained in 2 different batches of analyses should not be pooled together to fit

145 a single isochron, since they were corrected using different $k_{wm}$ and $l$. Instead, a weighted mean of the slopes obtained for each batch analyses can be calculated using Eq. (17):

$$b_{wm} = \frac{b_1\omega_1 + b_2\omega_2}{\omega_1 + \omega_2}, \tag{17}$$

where $\omega_i = \sigma^{-2}_{b_i int}$ .

Its internal uncertainty is given by Eq. (18):

150 $\sigma^2_{b_{wm}int} = \frac{1}{\omega_1 + \omega_2}$ . $\tag{18}$

Its external uncertainty is given by Eq. (19):

$$\sigma^2_{b_{wm}ext} = \sigma^2_{b_{wm}ext} \left(\frac{\omega_1}{\omega_1 + \omega_2}\right)^2 + \sigma^2_{b_{wm}ext} \left(\frac{\omega_2}{\omega_1 + \omega_2}\right)^2 + 2\sigma_{b_1,b_2} \left(\frac{\omega_1\omega_2}{(\omega_1 + \omega_2)^2}\right), \tag{19}$$

where $\sigma_{b_1,b_2}$ is given by Eq. (20):

$$\sigma_{b_1,b_2} = \sigma^2_\lambda \frac{b_1 b_2 t^2}{(e^{-\lambda t}-1)^2} + \sigma^2_t \frac{b_1 b_2 \lambda^2}{(e^{-\lambda t}-1)^2} + 2\sigma_{\lambda,t} \frac{b_1 b_2 t\lambda}{(e^{-\lambda t}-1)^2} + \sigma^2_{Y_0} \frac{b_1 b_2}{k_{wm1}k_{wm2}} \left(\frac{-1}{(e^{\lambda t}-1)\sum_1^{N1} w_{i1}} \sum_1^{N1} \frac{w_{i1}}{x_{i1}}\right)\left(\frac{-1}{(e^{\lambda t}-1)\sum_1^{N2} w_{i2}} \sum_1^{N2} \frac{w_{i2}}{x_{i2}}\right) +$$

155 $$+\sigma_{Y_0,t} \frac{b_1 b_2}{k_{wm1}k_{wm2}} \left(\left(\frac{-1}{(e^{\lambda t}-1)\sum_1^{N1} w_{i1}} \sum_1^{N1} \frac{w_{i1}}{x_{i1}}\right)\frac{\lambda k_{wm2}}{e^{-\lambda t}-1} + \frac{\lambda k_{wm1}}{e^{-\lambda t}-1}\left(\frac{-1}{(e^{\lambda t}-1)\sum_1^{N2} w_{i2}} \sum_1^{N2} \frac{w_{i2}}{x_{i2}}\right)\right) + \sigma_{l_1,k_{wm2}} \frac{-b_1 b_2}{l_1 k_{wm2}} +$$

$$+\sigma_{k_{wm1},l_2} \frac{-b_1 b_2}{k_{wm1}l_2} + \sigma_{l_1,l_2} \left(\frac{b_1 b_2}{l_1 l_2} + \frac{b_1 b_2}{k_{wm1}k_{wm2}}\left(\frac{k_{wm1}}{l_1} + \frac{Y_0}{l_1(e^{\lambda t}-1)\sum_1^{N1} w_{i1}} \sum_1^{N1} \frac{w_{i1}}{x_{i1}}\right)\left(\frac{k_{wm2}}{l_2} + \frac{Y_0}{l_2(e^{\lambda t}-1)\sum_1^{N2} w_{i2}} \sum_1^{N2} \frac{w_{i2}}{x_{i2}}\right)\right). \tag{20}$$

The multi-spot isochron date $T_{isochron}$ can be calculated from any of the above estimates for $b$ using Eq. (23):

$$T_{isochron} = \frac{\ln(b+1)}{\lambda}. \tag{21}$$

The uncertainty of $T_{isochron}$ is given by Eq. (25):

160 $\sigma^2_{T_{isochron}} = \sigma^2_b \left(\frac{1}{\lambda(b+1)}\right)^2 + \sigma^2_\lambda \left(\frac{-T_{isochron}}{\lambda}\right)^2,$ $\tag{22}$

where using the first term with $\sigma_{b\ int}$ provides the internal uncertainty ($\sigma_{T_{isochron\ int}}$), while using the entire equation with $\sigma_{b\ ext}$ provides the external uncertainty ($\sigma_{T_{isochron\ ext}}$).

## 2.2 Inverse Isochron Space

Following the same logic and assumptions to derive expressions for the inverse isochron space yields Eq. (1'-22'; numeration is preserved to facilitate correlation with the equations above and comments to those equations):

$$Y' = Y_0' + X'Y_0'(1 - e^{\lambda t}) , \tag{1'}$$

$$l'y' = Y_0' + k'x'Y_0'(1 - e^{\lambda t}) , \tag{2'}$$

$$k' = \frac{y'l' - Y_0'}{x'Y_0'(1 - e^{\lambda t})} , \tag{3'}$$

$$\sigma_{k'}^2 = \sigma_{x'}^2 \left(\frac{-k'}{x'}\right)^2 + \sigma_{y'}^2 \left(\frac{l'k'}{y'l' - Y_0'}\right)^2 + 2\sigma_{x',y'} \left(\frac{-l'k'^2}{x'(y'l' - Y_0')}\right) + \sigma_\lambda^2 \left(\frac{tk'}{e^{-\lambda t} - 1}\right)^2 + \sigma_t^2 \left(\frac{\lambda k'}{e^{-\lambda t} - 1}\right)^2 +$$

$$+ 2\sigma_{\lambda,t} \left(\frac{\lambda tk'^2}{(e^{-\lambda t} - 1)^2}\right) + \sigma_{l'}^2 \left(\frac{y'k'}{y'l' - Y_0'}\right)^2 + \sigma_{Y_0'}^2 \left(\frac{-y'l'k'}{Y_0'(y'l' - Y_0')}\right)^2 + 2\sigma_{Y_0',t} \left(\frac{-\lambda y'l'k'^2}{Y_0'(y'l' - Y_0')(e^{-\lambda t} - 1)}\right) , \tag{4'}$$

$$\sigma_{\lambda,t} = -\frac{t\sigma_\lambda^2}{\lambda} , \tag{5'}$$

$$\sigma_{Y_0',t} = -\frac{\sigma_{Y_0,t}}{Y_0^2} = \sigma_{Y_0'}^2 \frac{1 + X'^*(1 - e^{\lambda t})}{\lambda X'^* Y_0' e^{\lambda t}} = \sigma_{Y_0'}^2 \frac{1 - e^{\lambda t} - \eta'^*}{\lambda Y_0' e^{\lambda t}} , \tag{6'}$$

$$k'_{wm} = \frac{\sum_1^N k_i' w_i'}{\sum_1^N w_i'} , \tag{7'}$$

$$w_i' = \sigma_{k_i'\ int}^{-2} , \tag{7'a}$$

$$\sigma_{k'_{wm}}^2 = \frac{1}{\sum_1^N w_i'} + \sigma_\lambda^2 \left(\frac{tk'_{wm}}{e^{-\lambda t} - 1}\right)^2 + \sigma_t^2 \left(\frac{\lambda k'_{wm}}{e^{-\lambda t} - 1}\right)^2 + 2\sigma_{\lambda,t} \left(\frac{t\lambda k'^2_{wm}}{(e^{-\lambda t} - 1)^2}\right) + \sigma_{l'}^2 \left(\frac{1}{Y_0'(1 - e^{\lambda t})\sum_1^N w_i'} \sum_1^N \frac{y_i' w_i'}{x_i'}\right)^2 +$$

$$+ \sigma_{Y_0'}^2 \left(\frac{-l'}{Y_0'^2(1 - e^{\lambda t})\sum_1^N w_i'} \sum_1^N \frac{y_i' w_i'}{x_i'}\right)^2 + 2\sigma_{Y_0',t} \left(\frac{-\lambda k'_{wm} l'}{Y_0'^2(e^{\lambda t} + e^{-\lambda t} - 2)\sum_1^N w_i'} \sum_1^N \frac{y_i' w_i'}{x_i'}\right) , \tag{8'}$$

$$X_u' = k'_{wm} x_u' , \tag{9'}$$

$$Y_u' = l'y_u' , \tag{10'}$$

$$\sigma_{X_u'}^2 = \sigma_{x_u'}^2 k'^2_{wm} + \sigma_{k'_{wm}}^2 x_u'^2 , \tag{11'}$$

$$\sigma_{Y_u'}^2 = \sigma_{y_u'}^2 l'^2 + \sigma_{l'}^2 y_u'^2 , \tag{12'}$$

$$\sigma_{X_u',Y_u'} = \sigma_{x_u',y_u'} l'k'_{wm} + \sigma_l^2 y_u x_u \left(\frac{1}{Y_0'(1 - e^{\lambda t})\sum_1^N w_i'} \sum_1^N \frac{y_i' w_i'}{x_i'}\right) , \tag{13'}$$

$$T'_{spot} = \frac{\ln\left(1 - \frac{Y'_u - Y'_{0u}}{X'_u Y'_{0u}}\right)}{\lambda} \,, \tag{14'}$$

$$\sigma^2_{T'_{spot}} = \sigma^2_{X'_u}\left(\frac{Y'_u - Y'_{0u}}{\lambda X'_u(X'_u Y'_{0u} - Y'_u + Y'_{0u})}\right)^2 + \sigma^2_{Y'_u}\left(\frac{-1}{\lambda(X'_u Y'_{0u} - Y'_u + Y'_{0u})}\right)^2 + 2\sigma_{X'_u,Y'_u}\left(\frac{-(Y'_u - Y'_{0u})}{X'_u \lambda^2(X'_u Y'_{0u} - Y'_u + Y'_{0u})^2}\right) +$$

$$+ \sigma^2_{Y'_{0u}}\left(\frac{Y'_u}{\lambda Y'_{0u}(X'_u Y'_{0u} - Y'_u + Y'_{0u})}\right)^2 + \sigma^2_\lambda\left(\frac{-T_{spot}}{\lambda}\right)^2 \,, \tag{15'}$$


$$\sigma^2_{b'\,ext} = \sigma^2_{b'\,int} + \sigma^2_{k'_{wm}}\left(\frac{-b'}{k'_{wm}}\right)^2 + \sigma^2_{l'}\left(\left(\frac{b'}{l'}\right)^2 - \frac{2b'^2}{l' k'_{wm} Y'_0(1 - e^{\lambda t})\sum_1^N w'_i}\sum_1^N \frac{y'_i w'_i}{x'_i}\right) \,, \tag{16'}$$

$$b'_{wm} = \frac{b'_1 \omega'_1 + b'_2 \omega'_2}{\omega'_1 + \omega'_2} \,, \tag{17'}$$

$$\omega'_i = \sigma^{-2}_{b'_i int} \,, \tag{17'a}$$

$$\sigma^2_{b'_{wm} int} = \frac{1}{\omega'_1 + \omega'_2} \,, \tag{18'}$$

$$\sigma^2_{b'_{wm} ext} = \sigma^2_{b'_{wm} ext}\left(\frac{\omega'_1}{\omega'_1 + \omega'_2}\right)^2 + \sigma^2_{b'_{wm} ext}\left(\frac{\omega'_2}{\omega'_1 + \omega'_2}\right)^2 + 2\sigma_{b'_1,b'_2}\left(\frac{\omega'_1 \omega'_2}{(\omega'_1 + \omega'_2)^2}\right) \,, \tag{19'}$$


$$\sigma_{b'_1,b'_2} = \sigma^2_\lambda \frac{b'_1 b'_2 t^2}{(e^{-\lambda t} - 1)^2} + \sigma^2_t \frac{b'_1 b'_2 \lambda^2}{(e^{-\lambda t} - 1)^2} + 2\sigma_{\lambda,t}\frac{b'_1 b'_2 t\lambda}{(e^{-\lambda t} - 1)^2} +$$

$$+ \sigma^2_{Y_0} \frac{b'_1 b'_2}{k'_{wm1} k'_{wm2}}\left(\frac{-l'_1}{Y'^2_0(1 - e^{\lambda t})\sum_1^{N1} w'_{i1}}\sum_1^{N1}\frac{y'_{i1} w'_{i1}}{x'_i}\right)\left(\frac{-l'_2}{Y'^2_0(1 - e^{\lambda t})\sum_1^{N2} w'_i}\sum_1^{N2}\frac{y'_{1i} w'_{1i}}{x'_{1i}}\right) +$$

$$+ \sigma_{Y_0,t}\frac{b'_1 b'_2}{k'_{wm1} k'_{wm2}}\left(\left(\frac{-l'_1}{Y'^2_0(1 - e^{\lambda t})\sum_1^{N1} w'_{i1}}\sum_1^{N1}\frac{y'_{i1} w'_{i1}}{x'_{i1}}\right)\frac{\lambda k'_{wm2}}{e^{-\lambda t} - 1} + \frac{\lambda k'_{wm1}}{e^{-\lambda t} - 1}\left(\frac{-l'_2}{Y'^2_0(1 - e^{\lambda t})\sum_1^{N2} w'_{i2}}\sum_1^{N2}\frac{y'_{i2} w'_{i2}}{x'_{i2}}\right)\right) + \sigma_{l'_1,k'_{wm2}}\frac{-b'_1 b'_2}{l'_1 k'_{wm2}} +$$

$$+ \sigma_{k'_{wm1},l'_2}\frac{-b'_1 b'_2}{k'_{wm1} l'_2} + \sigma_{l'_1,l'_2}\left(\frac{b'_1 b'_2}{l'_1 l'_2} + \frac{b'_1 b'_2}{k'_{wm1} k'_{wm2}}\left(\frac{1}{Y'_0(1 - e^{\lambda t})\sum_1^{N1} w'_{i1}}\sum_1^{N1}\frac{y'_{i1} w'_{i1}}{x'_{i1}}\right)\left(\frac{1}{Y'_0(1 - e^{\lambda t})\sum_1^{N2} w'_{i2}}\sum_1^{N2}\frac{y'_{i2} w'_{i2}}{x'_{i2}}\right)\right) \,, \tag{20'}$$

$$T'_{isochron} = \frac{\ln\left(1 - \frac{b'}{Y'_{0u}}\right)}{\lambda} \,, \tag{21'}$$


$$\sigma^2_{T'_{isochron}} = \sigma^2_{b'}\left(\frac{-1}{\lambda(Y'_{0u} - b')}\right)^2 + \sigma^2_{Y'_{0u}}\left(\frac{b'}{Y'_{0u}\lambda(Y'_{0u} - b')}\right)^2 + 2\sigma_{Y'_{0u},b'}\left(\frac{-b'}{Y'_{0u}\lambda^2(Y'_{0u} - b')^2}\right) + \sigma^2_\lambda\left(\frac{-T'_{isochron}}{\lambda}\right)^2 \,, \tag{22'}$$

$$\sigma_{Y'_{0u},b'} = \sigma^2_{Y'_{0u}}\eta'_u \,, \tag{22'a}$$

where $Y'$ and $y'$ ($Y'_u$ and $y_u$) are the true and measured common to daughter isotope ratios (e.g., $^{86}Sr/^{87}Sr$) in the standard (unknown), which are related to each other via the mass dependent fractionation correction factor $l'$, $Y'_0$ ($Y'_{0u}$) is the initial common to daughter isotope ratio (e.g., $^{86}Sr/^{87}Sr_0$) in the standard (unknown), $X'$ and $x'$ ($X'_u$ and $x'_u$) are the true and measured parent to daughter ratios (e.g., $^{87}Rb/^{87}Sr$) in the standard (unknown), which are related to each other via the elemental

fractionation correction factor $k'$, $X'^*$ is the parent to daughter isotope ratio used to determine $t$ if it was determined from a single analysis, and $\eta'^*$ ($\eta'_u$) is the partial derivative of the isochron slope used to determine $t$ ($T'_{isochron}$) with respect to $Y'_0$ if it was determined by fitting an isochron.

## 2.3 Further considerations

I have tested that the above equations to estimate uncertainties perform as intended by comparing the estimates they yield for synthetic data with analogous estimates obtained using the Monte Carlo method. It thus should be possible to apply them in practice. However, it should be noted that it is not uncommon in practice to see greater dispersion in LA-ICP-MS data than predicted from theoretical considerations (Horstwood et al., 2016). I invite readers to consult Horstwood et al. (2016) on how to deal with this problem and also compare different sets of data. I would only highlight that when comparing sets of data from

different laboratories or publications one should consider whether they were obtained using the same standards and/or decay constants. If so, some covariance between dates in these sets is expected, and they should rather be compared using 'partial' external uncertainties that only account for uncertainties in those parameters that do not match.

## 3 Conclusion

The above equations can be used to first calculate elemental fractionation correction factors and their uncertainties from

individual analyses of primary standards with variable contents of parent and daughter isotopes, and then calculate isochron dates for individual or multiple analyses of unknowns and their uncertainties. Although it is yet to be tested how well the outlined approach performs in practice, it has two potential benefits over using non-matrix-matched primary standards and nanoparticulate pressed powder tablets as substitutes for matrix-matched primary standards. Firstly, it could be more suitable to characterise elemental fractionation in unknowns by providing better matrix matching. Secondly, it could reduce time

needed to analyse one batch of unknowns due to spending less time on acquiring data form primary standards that do not provide optimal matrix matching.

## Appendix

The following outlines how to estimate $\eta$, which is the partial derivative of the isochron slope with respect to $Y_0$, if the isochron was fitted by the method of York et al. (2004). I assume that numbering starts with 0, such that the 0-th term corresponds to

the initial composition $Y_0$, $X_0$ (normally $X_0 = 0$), and I use the notation of York et al. (2004) with the addition of $\eta$, $\Psi$ and $\Omega$. To calculate $\eta$ use Eq. (A1):

$$\eta = \frac{\Psi(\sum W_i \beta_i U_i) - \Omega(\sum W_i \beta_i V_i)}{(\sum W_i \beta_i U_i)^2} \; , \tag{A1}$$

where $\Psi$ and $\Omega$ are calculated using Eq. (A2-A3):

$$\Psi = W_0^2(X_0 - \bar{X})\left(\frac{1}{\omega(Y_0)} - \frac{br_0}{\alpha_0}\right) + 2W_0^2(Y_0 - \bar{Y})\left(\frac{b}{\omega(X_0)} - \frac{r_0}{\alpha_0}\right) -$$

$$-\frac{W_0}{\sum W_i}\sum W_i^2(X_i - \bar{X})\left(\frac{1}{\omega(Y_i)} - \frac{br_i}{\alpha_i}\right) - 2\frac{W_0}{\sum W_i}\sum W_i^2(Y_i - \bar{Y})\left(\frac{b}{\omega(X_i)} - \frac{r_i}{\alpha_i}\right), \quad \text{(A2)}$$

$$\Omega = W_0^2(X_0 - \bar{X})\left(\frac{b}{\omega(X_0)} - \frac{r_0}{\alpha_0}\right) + \frac{W_0}{\sum W_i}\sum W_i^2(X_i - \bar{X})\left(\frac{b}{\omega(X_i)} - \frac{r_i}{\alpha_i}\right). \quad \text{(A3)}$$

### Acknowledgements

I am grateful to Daniela Rubatto, Pieter Vermeesch and one anonymous reviewer for their comments and suggestions. While writing this manuscript I was supported by the Swiss National Science Foundation through Early Postdoc.Mobility (P2GEP2_191478) and Postdoc.Mobility (P500PN_202872) fellowships.

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

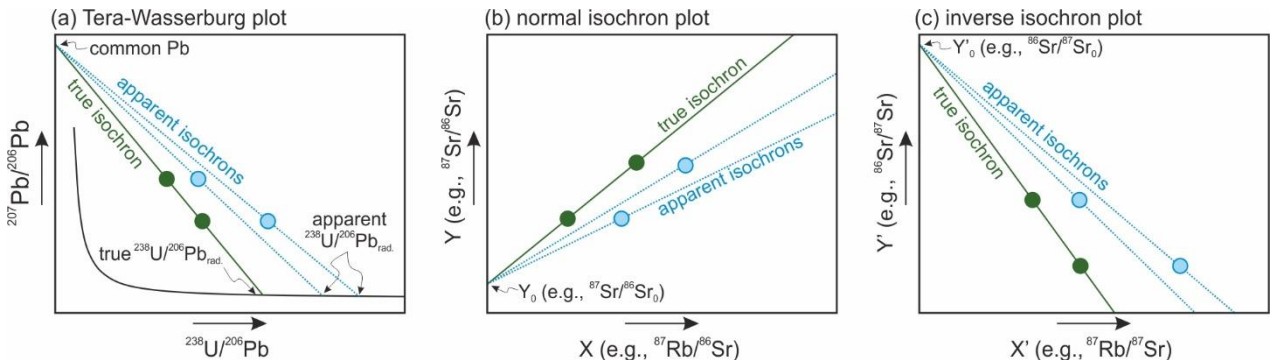

**Figure 1: Schematic illustrations for how individual analyses of primary standards with variable parent and daughter isotope concentrations can be used to obtain factors for elemental fractionation correction. Each plot shows two data points that are assumed to be corrected for mass dependent fractionation and have different elemental fractionation factors (e.g., due to instrument instability). (a) One of the approaches taken in U-Pb geochronological applications of LA-ICP-MS. Factors for U-Pb fractionation correction are calculated by rationing the true and apparent $^{238}$U/$^{206}$Pb$_{radiogenic}$ ratios that are obtained using the Tera-Wasserburg diagram. (b-c) Potential approaches for non-U-Pb geochronological applications of LA-ICP-MS. Factors for elemental fractionation correction can be estimated by finding coefficients by which the measured elemental ratios need to be multiplied to equate the slopes of the apparent and true isochrons, whether normal or inverse. Elemental fractionation correction factors can also be estimated by comparing the true and apparent intercepts with the horizontal axis in the inverse isochron diagram.**