# Peer review of "Short communication: On the potential use of materials with heterogeneously distributed parent and daughter isotopes as primary standards for non-U-Pb geochronological applications of laser ablation inductively coupled plasma mass spectrometry (LA-ICP-MS)"

_Geochronology, 2021_

## Referee Comment (RC2)

**Review of "On the potential use of materials with heterogeneously distributed parent and daughter isotopes ..." by D.V. Popov**

Pieter Vermeesch
University College London
`p.vermeesch@ucl.ac.uk`

January 7, 2022

I enjoyed reading this paper, which formalises a method to calibrate ICP-MS data using primary standards that contain a variable mixture of known radiogenic and inherited endmember compositions. The manuscript is well written and the logic is easy to follow. Nevertheless, I believe that the paper could be further improved by using a different statistical approach.

The method is most easily understood in conventional isochron space, which sets out two ratios $P/d$ and $D/d$, where $P$ and $D$ are the parent and daughter nuclides, and $d$ is a non-radiogenic isotope of the daughter element. Elemental fractionation only affects $P/d$ and not $D/d$. The correction algorithm assumes that the elemental fractionation can be captured by a single parameter, $k$ which, when multiplied with the measured $P/d$-ratios, brings them into alignment with an isochron of known intercept $(D/d)_0$ and slope $(e^{\lambda t} - 1)$.

In its present form, the algorithm estimates $k$ for each aliquot separately, and then averages these estimates into a single consensus value. Error propagation is done by conventional first order Taylor approximation. On lines 109–111 of the manuscript, the author gets stuck when trying to keep track of the systematic uncertainties ("I could not find an exact equation to calculate $a$", "$a$ could *probably* be estimated as a simple average").

I think that these issues can be solved with a different approach, using the method of Maximum Likelihood. The same method has been successfully applied in Ludwig (1998)'s seminal paper on U–Pb concordance, and I frequently use it in my own work (e.g. Vermeesch, 2020, for a recent example). In the next paragraphs, I will follow Ludwig (1998) and use uppercase symbols for measured quantities and lower case symbols for true (but unknown) values. Note that this is the opposite convention of Dr. Popov's manuscript. Let $x_i$ and $y_i$ be the true $P/d$- and $D/d$-ratios of the $i^{\text{th}}$ aliquot. Then $x_i$ and $y_i$ form an isochron:

$$y_i = y_0 + \left(e^{\lambda t} - 1\right) x_i \tag{1}$$

where $y_0$ is the non-radiogenic endmember composition of the primary standard, $t$ is its age and $\lambda$ is the decay constant. $x_i$ and $y_i$ are unknown, whereas $y_0$, $\lambda$ and $t$ are known within some uncertainty. $x_i$ and $y_i$ are related to the measurements $X_i$ and $Y_i$ as follows:

$$\begin{cases} X_i = kx_i + \epsilon(X_i) \\ Y_i = y_i + \epsilon(Y_i) \end{cases} \tag{2}$$

where $k$ is the elemental fractionation factor and $\epsilon(X_i)$ and $\epsilon(Y_i)$ are bivariate normal residuals. We will assume that these are adequately captured by the measurement uncertainties propagated from the mass spectrometer data. I will now outline two algorithms to estimate $k$, first without and then with the systematic uncertainties of $y_0$, $\lambda$ and $t$.

1. Without systematic uncertainties:

Define the log-likelihood $\mathcal{LL}$ of the data given $k$ and $x = \{x_1, \ldots, x_i, \ldots, x_n\}$:

$$\mathcal{LL} = \sum_{i=1}^{n} \Delta_i^T \Omega_i \Delta_i \tag{3}$$

where $n$ is the number of aliquots for the primary standard,

$$\Delta_i = \left[ \begin{array}{c} kx_i - X_i \\ y_0 + (e^{\lambda t} - 1)x_i - Y_i \end{array} \right], \tag{4}$$

$$\Omega_i = \left[ \begin{array}{cc} \sigma[X_i]^2 & \sigma[X_i, Y_i] \\ \sigma[X_i, Y_i] & \sigma[Y_i]^2 \end{array} \right]^{-1} \tag{5}$$

and $\Delta^T$ is the transpose of $\Delta_i$.

Then $k$ can be estimated by maximising $\mathcal{LL}$ with respect to $k$ and $x_1 \ldots x_n$. I lack the time to work out the details, but it should be possible to do this by taking the first derivative w.r.t. the $x_i$s and setting it to zero, followed by numerical optimisation for $k$. See Ludwig (1998) for an example.

2. With systematic uncertainties:

Instead of the sum of $n$ terms shown in Equation 3, the maximum likelihood estimation can also be captured in a single matrix expression. This has the benefit that it allows the uncertainties of $y_0$, $\lambda$ and $t$ to be captured in the estimation of $k$:

$$\mathcal{LL}' = \Delta'^T \left( J \Sigma_i J^T \right)^{-1} \Delta' \tag{6}$$

where

$$\Delta' = \left[ \begin{array}{c} kx_1 - X_1 \\ \vdots \\ kx_n - X_n \\ y_0 + \left(e^{\lambda t} - 1\right) x_1 - Y_1 \\ \vdots \\ y_0 + \left(e^{\lambda t} - 1\right) x_n - Y_n \end{array} \right], \tag{7}$$

$$\Sigma = \left[ \begin{array}{ccccccccc} \sigma[X_1]^2 & \cdots & 0 & \sigma[X_1, Y_1] & \cdots & 0 & 0 & 0 & 0 \\ \vdots & \ddots & \vdots & \vdots & \ddots & \vdots & \vdots & \vdots & \vdots \\ 0 & \cdots & \sigma[X_n]^2 & 0 & \cdots & \sigma[X_n, Y_n] & 0 & 0 & 0 \\ \sigma[X_1, Y_1] & \cdots & 0 & \sigma[Y_1]^2 & \cdots & 0 & 0 & 0 & 0 \\ \vdots & \ddots & \vdots & \vdots & \ddots & \vdots & \vdots & \vdots & \vdots \\ 0 & \cdots & \sigma[X_n, Y_n] & 0 & \cdots & \sigma[Y_n]^2 & 0 & 0 & 0 \\ 0 & 0 & 0 & 0 & 0 & 0 & \sigma[y_0]^2 & \sigma[y_0, t] & \sigma[y_0, \lambda] \\ 0 & 0 & 0 & 0 & 0 & 0 & \sigma[y_0, t] & \sigma[t]^2 & \sigma[t, \lambda] \\ 0 & 0 & 0 & 0 & 0 & 0 & \sigma[y_0, \lambda] & \sigma[t, \lambda] & \sigma[\lambda]^2 \end{array} \right] \tag{8}$$

and

$$J = \left[ \begin{array}{ccccc} -I_{n,n} & 0_{n,n} & 0_{n,1} & 0_{n,1} & 0_{n,1} \\ 0_{n,n} & -I_{n,n} & 1_{n,1} & \lambda e^{\lambda t} x & t e^{\lambda t} x \end{array} \right] \tag{9}$$

where $I_{a,b}$ is the $a \times b$ identity matrix, $0_{a,b}$ is an $a \times b$ matrix of zeros, and $x$ is the $n$-element column vector of $x_i$-values. As before, the $k$ and $x_i$-values are estimated by maximising $\mathcal{LL}'$, which can be done using numerical methods.

According to maximum likelihood theory, the covariance matrix of the estimated parameters can be obtained by inverting the matrix of second derivatives of the log-likelihood in the vicinity of the maximum. This is the approach used by Ludwig (1998) and it can also be applied to the present problem. If the author is not familiar with this method, then a simple proof is provided in Section 8.4 of these lecture notes: `https://github.com/pvermees/geostats/blob/main/latex/geostats.pdf`

I am happy to answer any questions arising from my review via the email address provided above.

**References**

Ludwig, K. R. On the treatment of concordant uranium-lead ages. *Geochimica et Cosmochimica Acta*, 62: 665–676, 1998.

Vermeesch, P. Unifying the U–Pb and Th–Pb methods: joint isochron regression and common Pb correction. *Geochronology*, 2(1):119–131, 2020.

---

## Author Comment (AC1)

Equations 15 and 16 in the manuscript should read as follows:

$$a = \sigma_{Y_0}^2 \left( \frac{-1}{e^{\lambda t}-1} \left( \frac{1}{Nx_1} + \cdots + \frac{1}{Nx_N} \right) \right)^2 + \frac{2\sigma_{Y_0,t}\lambda k_{av}}{(e^{\lambda t}+e^{-\lambda t}-2)} \left( \frac{1}{Nx_1} + \cdots + \frac{1}{Nx_N} \right), \tag{15}$$

$$a = \sigma_{Y_0'}^2 \left( \frac{-1}{Y_0'^2(1-e^{\lambda t})} \left( \frac{y_1'}{Nx_1'} + \cdots + \frac{y_N'}{Nx_N'} \right) \right)^2 + \frac{2\sigma_{Y_0',t}\lambda k_{av}}{Y_0'^2(2-e^{\lambda t}-e^{-\lambda t})} \left( \frac{y_1'}{Nx_1'} + \cdots + \frac{y_N'}{Nx_N'} \right), \tag{16}$$

Equation 12 in the manuscript should read as follows:

$$\sigma_{Y_0',t} = -\frac{\sigma_{Y_0,t}}{Y_0^2} = \sigma_{Y_0'}^2 \frac{1+X'^*(1-e^{\lambda t})}{\lambda X'^* Y_0' e^{\lambda t}}, \tag{12}$$

---

## Author Comment (AC2)

This short communication is a technique-based manuscript, useful for those performing LA-ICPMS dating for systems other than U-Pb—that is, those with only one parent/daughter —that also have variable parent and daughter concentrations. It includes a standardization technique for correcting raw parent/daughter ratios, subject to elemental fractionation by laser ablation, transport, ionization efficiency, etc.. The general idea, as follows, is no different than correction of LA-ICPMS U-Pb data, which has been explored by many of the authors referenced within: 1) correct for mass bias of the daughter ratio (can be done a number of ways, including the use of a non-matrixed matched RM (reference material), via solution, or internal standardization of a non-U-Pb system) and correct all RMs and unknowns accordingly; 2) assume concordance for the RM and correct the parent/daughter ratio, such that the age matches it's accepted value. This is a relatively straightforward correction that has been explained many times over, primarily for U-Pb. As such, this communication seems a touch superfluous, as a single isotopic geochronometer is simpler than the U-Pb system, but nevertheless is rarely mentioned and therefore warrants more discussion, especially in the light of recent developments in LA-ICP dating techniques (e.g., Zack and Hogmalm, 2016 and Simpson et al., 2021).

In my experience, the best example of standardization of elemental fractionation of common-daughter-bearing minerals is that in Chew et al., 2014, and I shall thus refer to it often below; though the Chew et al. study discusses the U-Pb system, it does so on a system-by-system basis, that is, it corrects 206Pb/238U and 207Pb/235U ratios using any of the other isotopes of the daughter product of the system (i.e., 204¬Pb, 207¬Pb, 208Pb for 206Pb/238U and 204¬Pb, 206¬Pb, 208Pb for 207Pb/235U). As an example, one can look at Fig. 2E, in which each parent/daughter ratio has been corrected using a non-radiogenic daughter (204¬Pb); the math by which to do this should be identical to the math by which to correct any spot analysis for any radioisotopic system - that is, it is should be identical to Equation 21 in this manuscript. Nevertheless, it is not spelled out in this paper at least, that the calculation for U-Pb applies the same way for other isotopic systems such as Rb-Sr, Sm-Nd, Lu-Hf etc., which is presumably why the author has endeavored to write this short communication.

The approach with the [204]Pb-based correction method should be equivalent to that outlined in my manuscript, although the math, at least in UcomPbine, is not exactly the same (if I understand correctly the IgorPro language in UcomPbine file that I have). This similarity escaped my attention before, partly because the [207]Pb-based correction method is used more widely, but I can mention it in the revised manuscript. Another notable similarity, is that with the [40]Ar/[39]Ar method: division by factor k in Eq. 21, 24, 26 and 28 is similar to multiplication by factor J in the [40]Ar/[39]Ar method.

What the Chew et al. study doesn't explain as well is how to correct the mass bias for the ratio of the daughter isotopes (e.g., 207Pb/206¬Pb, 207Pb/204¬Pb, 87Sr/86¬Sr, etc.). Unfortunately, that is also mostly missing from this manuscript, which should be revised to state how this can/should be done in a clear and concise manner; for non-U-Pb LA-ICPMS geochronology—Rb/Sr, Sm/Nd, Lu/Hf—the mass fractionation (Y-axis value) can be calculated internally, unlike for U-Pb, which has no two non-radiogenic isotopes (however this internal standardization is rarely done - this needs discussion). The analytical uncertainty in this correction is likely to be in the 10's low 100's of ppm (<<1%) and for intents and purposes, can be considered negligible when calculating age uncertainties, however, the actual uncertainty of the measurement—because of interferences and matrix effects, for example—is likely to be much larger.

One of the reasons why I selected "short communication" as the article type is that I did not want to go into these details. I am not the best person to advise on this subject, and I also do not have time to thoroughly review relevant literature in the near future. However, I can show how to propagate the uncertainties related to the mass fractionation correction to the date uncertainties. Unfortunately, I currently do not have any real-world data to assess the importance of this. According to my brief tests with synthetic data, % level of uncertainty for the mass fractionation correction factor are needed to have a significant impact on the date uncertainty, which seems to be unrealistically high.

Notably, it is possible to completely ignore the mass and fractionation correction if (i) the primary standard is sufficiently heterogeneous in terms of the parent to daughter isotope ratio, (ii) there is no significant instrument drift so that multiple primary standard measurements can be used to calculate $k$, (iii) only isotope ratios measured during one session of LA-ICP-MS analyses enter the equations to calculate the date of the unknown, (iv) the date of the unknown is calculated as a multi-spot isochron date. In these circumstances, the slope $b_s$ of the line fitted through the standard data can be used to calculate the factor $k$ and its uncertainty:

$$k = \frac{b_s}{e^{\lambda t}-1} \; ; \; \sigma_k^2 = \sigma_{b_s}^2 \left(\frac{k}{b_s}\right)^2 + \sigma_t^2 \left(\frac{-\lambda k}{1-e^{-\lambda t}}\right)^2 + \sigma_\lambda^2 \left(\frac{-tk}{1-e^{-\lambda t}}\right)^2 + 2\sigma_{\lambda,t} \left(\frac{t\lambda k^2}{(1-e^{-\lambda t})^2}\right).$$

The righthand side of the first of the above equations can be substituted for $k_{av}$ in Eq. 24 from the manuscript (the one needed to calculate the date of the unknown):

$$T_{isochron} = \frac{\ln\left(\frac{b}{b_s}(e^{\lambda t}-1)+1\right)}{\lambda}.$$

Considering that mass fractionation has the effect of multiplying both $b$ and $b_s$ by the same constant, the above equation indicates that it can be completely ignored (this constant will be cancelled out). The uncertainty of thereby calculated date can be estimated using Eq. 25 from the manuscript with appropriate substitutions for $k_{av}$ and $\sigma_{k_{av}}$. Note that the only parameter that is needed to characterise the primary standard in the described scenario is its date, and its use would be analogous to the use of the neutron flux monitors in the $^{40}Ar/^{39}Ar$ method.

On this note, these excess uncertainties are not included in the equations herein, as far as I can tell, and in many cases, these types of uncertainties are likely to be the biggest cause of the actual uncertainty of the measurement. One of the seminal papers in uncertainty propagation for LA-ICPMS dating is that of Horstwood et al., 2016, in which they explain how the reproducibility of measurements can easily overwhelm the instrument analytical uncertainty. In that paper, without equations, they give their best practices for data reduction workflow, which include propagating excess uncertainty (different than external uncertainty). This is a critical step in reporting ages and uncertainties in all LA-ICPMS derived data and cannot be ignored in the current manuscript.

I can show how to propagate the uncertainties associated with the mass fractionation correction to the date uncertainties. Additionally, the date uncertainty can be affected by the isobaric interference corrections. However, these effects would be specific for each method and setup, and therefore I cannot derive generic formulas. In theory, the addition of these two sources of uncertainty to the revised equations from my reply to Pieter Vermeesch's review should provide means to estimate the full external uncertainties in a way that is generally consistent with Horstwood et al. (2016). The way Horstwood et al. (2016) proposed to account for the excess uncertainties that become evident from the repeated analyses of secondary standards is, in my understanding, an ad hoc protocol for what to do, if the chosen approach to estimate the full external uncertainties demonstrably did not work. This problem goes beyond the scope of my work, and I suggest that I will simply refer readers to Horstwood et al. (2016) to see their recommendations.

The main aspect of this paper that is relevant, and has not been discussed in great detail, is the correction of parent/daughter ratios and consequent age calculation using a standard isochron method, that is, a graph in which both axes have a non-radiogenic, non-radioactive daughter isotope as the denominator (or numerator on the Y-axis in an inverse diagram; this is opposed to a Tera-Wasserburg diagram, for example, which uses radiogenic daughters on both axes). Again, the correction of the ratios for each axis (ratio) of this diagram have been described in numerous publications (primarily

for U-Pb, but see Zack and Hogmalm, 2016 and Simpson et al., 2021, and furthermore there is no difference in the correction method between that and non-U-Pb geochronometers), but few 1) demonstrate visually the uncorrected vs. corrected data, or 2) give the equations for uncertainties for each parameter. Point 1) is easy enough to do on one's own to get a visual representation of the 2-step correction for each ratio, and is analogous to the correction of U-Pb on a TW diagram as shown in Chew et al., 2014, Fig. A1. As noted above, this figure is missing the daughter-ratio correction, and would be more appropriate shown below, but this time in a single-system isochron diagram (analogous to Fig 1b in the submitted manuscript):

Most U-Pb applications that I am aware of use the [207]Pb-based correction method. I mention in the introduction that this method is similar. The [204]Pb-based method is equivalent to that outlined in my manuscript, but I cannot easily recall any studies that employ it to use common-Pb bearing standards (except for Chew et al., 2014, where it is one of the methods), and I certainly have not seen any studies that explain how to propagate uncertainties when using it. I can highlight the similarity in the revised manuscript. Zack and Hogmalm (2016) do something similar to what I describe in the manuscript, which I mention in the introduction. Simpson et al. (2021) do correct sample data for common Hf before correcting thereby calculated 176Hf$_r$/176Lu ratios for elemental fractionation. However, with some adjustments, these two corrections could be done in the reverse order. Overall, I am missing the point of the above paragraph.

Note that the figures in the current manuscript are either misleading or wrong. Given that there is little discussion about the correction of the y-axis, my impression is that it is the latter; the plots do not accurately represent theoretical data, as data of the same age, whether real or synthetic, should be isochronous, whether corrected for elemental fractionation or not. Given that the math for generating such apparent and corrected isochrons is trivial, it is worrisome that the plots in Figure 1 are incorrectly represented.

Each of these figures shows two data points that are assumed to be corrected for mass dependent fractionation and have different elemental fractionation factors (for example, due to instrument instability). The idea was to show that factors to correct for elemental fraction can be calculated from individual analyses, revealing any instrument drift over analytical sessions. I can clarify this in the revised manuscript.

In conclusion, for this manuscript to merit publication, it must first contain a broader background of previous work, and a better description of the workflow to correcting measured ratios, both for

elemental fractionation (including differences fractionation down-hole which is completely missing). Second, it needs a better description of all possible sources of uncertainty and how and when they should be properly propagated. Third, any figure must accurately represent real-world data.

I think that my introduction already makes a fair overview of previous work, and the only missing point is that about the similarity with the [204]Pb-based correction method applied to primary standards in U-Pb applications of LA-ICP-MS. I do not think that a short communication needs to make a thorough review of every aspect of LA-ICP-MS data treatment, and I was hoping to avoid this by choosing this article type. I also think that the intended readership will have a general knowledge of how to treat LA-ICP-MS data. I do not agree that a more detailed description of a workflow for data treatment is prerequisite. I presume that the comment requests to do something similar to Horstwood et al. (2016). However, I do not fully agree with that paper to just copy the outline, and I would like to avoid engaging in lengthy discussions and arguments that may arise during the review (hence I chose "short communication" as the article type). There is no error in my figure 1. I can adjust equations to include the uncertainties that are associated with the mass fractionation correction (see below).

**How to propagate the uncertainty related to the mass fractionation correction the date uncertainty in the normal isochron space**

Say the true value $y$ is obtained by multiplying the measured value $y_m$ by the constant $l$:

$$y = y_m l \, .$$

The uncertainty of $y$ can be calculated as follows:

$$\sigma_y^2 = \sigma_{y_m}^2 l^2 + \sigma_l^2 y_m^2 \, ,$$

where the first term provides the internal uncertainty, while the entire equation provides the external uncertainty. The entire Equation 4 from the manuscript should be used to calculate the external uncertainties of individual estimates for $k$, while their internal uncertainties should be calculated by using the first three terms and substituting $\sigma_{x_{int}}$ and $\sigma_{y_{int}}$ for $\sigma_x$ and $\sigma_y$. The uncertainty of the averaged value $k_{av}$ can be calculated by adding the term $c$ to Eq. 14 form the manuscript:

$$c = \frac{\sigma_l^2}{N^2 l^2 \left(e^{\lambda t}-1\right)^2} \left(\frac{y_1}{x_1} + \cdots + \frac{y_N}{x_N}\right)^2 \, .$$

The uncertainty of the spot date can be calculated by adding the term $d$ to Eq. 22 from the manuscript (the full external uncertainties $\sigma_{x_u}$ and $\sigma_{y_u}$ should be used in that equation):

$$d = 2\frac{\sigma_l^2}{l^2} y_u \left( k_{av} + \frac{Y_o}{(e^{\lambda t}-1)} \left( \frac{1}{Nx_1} + \cdots + \frac{1}{Nx_N} \right) \right) \frac{y_{0u}-y_u}{k_{av}\lambda^2(y_u-y_{0u}+k_{av}x_u)^2}.$$

The uncertainty of the multi-spot isochron date can be calculated by adding the term $f$ to Eq. 25 from the manuscript (the internal uncertainties $\sigma_{x_{u\,int}}$ and $\sigma_{y_{u\,int}}$ should be used to calculate the uncertainty $\sigma_b$ that should be plugged into this equation):

$$f = 2\frac{\sigma_l^2}{l^2} \frac{b^2}{\lambda^2(b+k_{av})^2} \left( -1 - \frac{2Y_o}{k_{av}(e^{\lambda t}-1)} \left( \frac{1}{Nx_1} + \cdots + \frac{1}{Nx_N} \right) \right).$$

---

## Author Response (AR1)

Dear Prof. Rubatto,

Thank you for handling my manuscript and providing thoughtful comments and suggestions. I have introduced some considerable changes to address the received comments:

1) Equations for normal and inverse isochron spaces were put in separate sections to facilitate reading and understanding, and notation was made more consistent.

2) Revised equations account for the uncertainty related to the mass fraction correction.

3) All arithmetic means are replaced with weighted means.

4) Intermediate steps were introduced that show how to calculate corrected compositions of unknowns with associated internal and external uncertainties. Hopefully, this will make it easier to understand how to apply these equations and prepare data tables for publication.

5) An approach to combine data from multiple analytical sessions to calculate a single isochron data was outlined.

6) The revised manuscript includes a better way to estimate the covariance between the age of the primary standard and its initial isotopic composition.

7) The revised manuscript mentions the approach with the $^{204}$Pb-based correction, cites one more non-U-Pb study that used a heterogeneous material to correct for elemental fractionation, and provides a citation for using 2 non-radiogenic isotopes to monitor mass dependent fractionation.

8) The work of Horstwood et al. (2016) was cited.

9) My affiliations and funding information were updated.

Kind regards,
Daniil

**Daniela Rubatto**

Dear Dr Popov

Thanks for providing replies to the comments of the referees.

Regarding the comment of Vermeesch, I accept that you will add to the paper your equation for calculating systematic uncertainties. Additionally, you could however mentionthat an alternative strategy would be that of Maximum Likelihood with reference to Ludwig work.

The revised manuscript includes equations that were modified in accordance with my previous reply and also mentions this alternative strategy with the reference to Pieter Vermeesch's review.

Regarding the comment on Referee 1:

- I agree that a short communication may have a lean introduction, but acknowledgement to previous studies is still due, even if without much discussion. Thus please addadditional reference in the introduction for the common Pb correction strategies.

The revised manuscript has the reference to the 204Pb-based correction method.

- Please add to your manuscript the explanation of how to propagate the uncertainties related to the mass fractionation correction.

The revised manuscript shows how to do this.

- The application of your method to "real world data" is recommended as it will improve understanding and strengthen the paper. The absence of real world data is surely not a difficult hurdle to overcome given that you are in a department where LA-ICPMS geochronology is well established. Contacting authors that have published LA-ICPMS dataset for Rb-Sr would be an alternative. I suggest you at least give it a try.

According to my tests with synthetic data, it should work. I have already left that department and exist in a very unstable situation that precludes any kind of collaboration to get hands on real-world data in the near future.

- Adding a workflow to follow for the correct implementation of your method and calculation of uncertainties is more critical and highly recommended. This is yes a short technical communication, but if you want others to adopt and properly use your method, a workflow give will go a long way in making it more accessible. Such a workflow requires not much discussion, particularly if adapted from Horstwood et al. 2016.

I rearranged formulas into two subsections (2.1 and 2.3), introduced some simplifications and intermediate steps and provided more detailed explanations, so it should be easier to follow the manuscript and use it as an outline for a workflow. I placed a reference to Horstwood et al. (2016) in subsection 2.3.

- Please clarify the meaning of figure 1 to avoid any miss interpretation. I have an additional comment for Fig. 1a. The line between age on Concordia and initial Pb on the Y-axis should not be named "Discordia" as this term is conventionally used forthe line on the Wetherill Concordia plots that joins two radiogenic ratios. For the TW diagram I suggest to use "regression". I am looking forward to receive the revised version of your manuscript. Kind regardsDaniela Rubatto

The revised figure caption provides more clarity. I changed 'discordias' to 'isochrons', which is what those lines essentially represent.

**Pieter Vermeesch**

I have derived appropriate formulas, which now have been incorporated in Eq. (8 and 8'). The introductory part of section 2 now includes a reference to the discussion of the maximum likelihood method in Pieter Vermeesch's review.

**Anonymous reviewer**

This short communication is a technique-based manuscript, useful for those performing LA-ICPMS dating for systems other than U-Pb—that is, those with only one parent/daughter —that also have variable parent and daughter concentrations. It includes a standardization technique for correcting raw parent/daughter ratios, subject to elemental fractionation by laser ablation, transport, ionization efficiency, etc.. The general idea, as follows, is no different than correction of LA-ICPMS U-Pb data, which has been explored by many of the authors referenced within: 1) correct for mass bias of the daughter ratio (can be done a number of ways, including the use of a non-matrixed matched RM (reference material), via solution, or internal standardization of a non-U-Pb system) and correct all RMs and unknowns accordingly; 2) assume concordance for the RM and correct the parent/daughter ratio, such that the age matches it's accepted value. This is a relatively straightforward correction that has been explained many times over, primarily for U-Pb. As such, this communication seems a touch superfluous, as a single isotopic geochronometer is simpler than the U-Pb system, but nevertheless is rarely mentioned and therefore warrants more discussion, especially in the light of recent developments in LA-ICP dating techniques (e.g., Zack and Hogmalm, 2016 and Simpson et al., 2021).

In my experience, the best example of standardization of elemental fractionation of common-daughter-bearing minerals is that in Chew et al., 2014, and I shall thus refer to it often below; though the Chew et al. study discusses the U-Pb system, it does so on a system-by-system basis, that is, it

corrects 206Pb/238U and 207Pb/235U  ratios using any of the other isotopes of the daughter product of the system (i.e., 204¬Pb, 207¬Pb, 208Pb for 206Pb/238U  and  204¬Pb, 206¬Pb, 208Pb for 207Pb/235U). As an example, one can look at Fig. 2E, in which each parent/daughter ratio has been corrected using a non-radiogenic daughter (204¬Pb); the math by which to do this should be identical to the math by which to correct any spot analysis for any radioisotopic system - that is, it is should be identical to Equation 21 in this manuscript. Nevertheless, it is not spelled out in this paper at least, that the calculation for U-Pb applies the same way for other isotopic systems such as Rb-Sr, Sm-Nd, Lu-Hf etc., which is presumably why the author has endeavored to write this short communication.

The revised manuscripts cites the approach with the [204]Pb-based correction.

What the Chew et al. study doesn't explain as well is how to correct the mass bias for the ratio of the daughter isotopes (e.g., 207Pb/206¬Pb, 207Pb/204¬Pb, 87Sr/86¬Sr, etc.). Unfortunately, that is also mostly missing from this manuscript, which should be revised to state how this can/should be done in a clear and concise manner; for non-U-Pb LA-ICPMS geochronology—Rb/Sr, Sm/Nd, Lu/Hf— the mass fractionation (Y-axis value) can be calculated internally, unlike for U-Pb, which has no two non-radiogenic isotopes (however this internal standardization is rarely done - this needs discussion). The analytical uncertainty in this correction is likely to be in the 10's low 100's of ppm (<<1%) and for intents and purposes, can be considered negligible when calculating age uncertainties, however, the actual uncertainty of the measurement—because of interferences and matrix effects, for example—is likely to be much larger.

The revised manuscript cites a paper that uses the suggested approach to calculate factors for the mass fractionation correction. It also shows how to account for the uncertainties in these correction factors. I think that this is sufficient for a "short communication" that is intended for those who are in a position to use the equations from it.

On this note, these excess uncertainties are not included in the equations herein, as far as I can tell, and in many cases, these types of uncertainties are likely to be the biggest cause of the actual uncertainty of the measurement. One of the seminal papers in uncertainty propagation for LA-ICPMS dating is that of Horstwood et al., 2016, in which they explain how the reproducibility of measurements can easily overwhelm the instrument analytical uncertainty. In that paper, without equations, they give their best practices for data reduction workflow, which include propagating excess uncertainty (different than external uncertainty). This is a critical step in reporting ages and uncertainties in all LA-ICPMS derived data and cannot be ignored in the current manuscript.

The revised manuscript mentions this problem and refers to Horstwood et al. (2016) to see their recommendations.

The main aspect of this paper that is relevant, and has not been discussed in great detail, is the correction of parent/daughter ratios and consequent age calculation using a standard isochron method, that is, a graph in which both axes have a non-radiogenic, non-radioactive daughter isotope as the denominator (or numerator on the Y-axis in an inverse diagram; this is opposed to a Tera-Wasserburg diagram, for example, which uses radiogenic daughters on both axes). Again, the correction of the ratios for each axis (ratio) of this diagram have been described in numerous publications (primarily for U-Pb, but see Zack and Hogmalm, 2016 and Simpson et al., 2021, and furthermore there is no difference in the correction method between that and non-U-Pb geochronometers), but few 1) demonstrate visually the uncorrected vs. corrected data, or 2) give the equations for uncertainties for each parameter. Point 1) is easy enough to do on one's own to get a visual representation of the 2-step correction for each ratio, and is analogous to the correction of U-Pb on a TW diagram as shown in Chew et al., 2014, Fig. A1. As noted above, this figure is missing the daughter-ratio correction, and would be more appropriate shown below, but this time in a single-system isochron diagram (analogous to Fig 1b in the submitted manuscript):

The revised manuscript mentions the approach with the $^{204}$Pb-based correction method. The work of Zack and Hogmalm (2016) was cited in the submitted version, and this citation remained in the revised manuscript. I have found and cited one more study that utilises a similar approach.

I did not cite Simpson et al. (2021). They do correct sample data for common Hf before correcting thereby calculated 176Hf$_r$/176Lu ratios for elemental fractionation, however, with some adjustments, these two corrections could be done in the reverse order.

Note that the figures in the current manuscript are either misleading or wrong. Given that there is little discussion about the correction of the y-axis, my impression is that it is the latter; the plots do not accurately represent theoretical data, as data of the same age, whether real or synthetic, should be isochronous, whether corrected for elemental fractionation or not. Given that the math for generating such apparent and corrected isochrons is trivial, it is worrisome that the plots in Figure 1 are incorrectly represented.

Each of these figures shows two data points that are assumed to be corrected for mass dependent fractionation and have different elemental fractionation factors (e.g. due to instrument instability). The idea was to show that factors to correct for elemental fraction can be calculated from individual analyses, revealing any instrument drift over analytical sessions. The revised figure caption explains this.

In conclusion, for this manuscript to merit publication, it must first contain a broader background of previous work, and a better description of the workflow to correcting measured ratios, both for elemental fractionation (including differences fractionation down-hole which is completely missing). Second, it needs a better description of all possible sources of uncertainty and how and when they should be properly propagated. Third, any figure must accurately represent real-world data.

I rearranged equations from the previous version, added some additional ones and provided more explanations to facilitate the implementation of the proposed approach. I think that this is sufficient for intended readership of my "short communication".

---

## Author Response (AR2)

Dear Prof. Rubatto,

Thank you for your comments and understanding regarding my situation. I have introduced the requested changes.

Kind regards,
Daniil Popov

**Daniela Rubatto**

Dear Dr Popov

Thanks for providing the revised version of you paper. Thanks for clarifying the error propagation procedure and for slightly improving the introduction. Although regrettable, I accept that it is currently impossible to get a real dataset to test your approach. I am satisfied that the manuscript has improved and is in principle acceptable. However there are still some minor corrections needed to clarify the language/nomenclature.

Line 45, "Apparently" is not the correct word here, better delete it.

I have rephrased this part. I also fixed the last sentence of this paragraph, which looked a bit out of context after the previous iteration.

In figure 1a, I insist that the nomenclature of the regression lines be corrected.

"isochrons" are replaced with "projections"

Line 73. In the main text do not refer to the reviewer's name directly. Simply state that alternative approaches exist and provide a supporting reference for it (e.g. Ludwig 1998 or Vermeesch 2020).

I have cited Pieter Vermeesch's review as if it was a paper. References to other papers in that review were to examples applying this statistical method to problems that are not directly related to my manuscript.

An isochron is a regression line in a plot where both isotopic ratios contain a non-radiogenic isotope (see also comments from reviewer) and where the slope of the line defines a unique age (as correctly

named in figure 1b). Both these conditions are not true for the regression in the TW diagrams (207/206 vs 238/206) of uncorrected ratios you use. Therefore I ask you to change the nomenclature to "regression" or another neutral term. I acknowledge that other publications use the term isochron for the TW regression, but this mistake should not be propagated.

I have changed it to "projections", although in my opinion it is legitimate to call these lines "isochrons" (they are conceptually very similar to inverse isochrons).

Bets regards
Daniela Rubatto